# The effect of photobiomodulation on histamine and Mucuna pruriens-induced pruritus, hyperknesis and alloknesis in healthy volunteers: A double-blind, randomized, sham-controlled study

Kordula Lang-Illievich[1,2☯], Christoph Klivinyi[1☯], Heike Schulze-Bauer[1], Ala Elhelali[3], Helmar Bornemann-Cimenti[1] *

1 Department of Anaesthesiology and Intensive Care Medicine, Medical University of Graz, Graz, Austria,
2 Department of Anesthesia and Intensive Care Medicine, Klinik Güssing, Güssing, Austria, 3 Department of Plastic and Reconstructive Surgery, Johns Hopkins University, Baltimore, Maryland, United States of America

☯ These authors contributed equally to this work.
* helmar.bornemann@medunigraz.at

**Data Availability Statement:** The data are available from https://osf.io/hfqju.

## Abstract

### Background

Photobiomodulation, also referred to as Low-Level Light Therapy (LLLT), has emerged as a promising intervention for pruritus, a prevalent and often distressing symptom.

### Objectives

This study investigated the efficacy of low-level light therapy (LLLT) in alleviating pruritus, hyperknesis, and alloknesis induced by histamine and Mucuna pruriens.

### Methods

In a double-blind, randomized, sham-controlled trial with a split-body design, healthy volunteers underwent 6 minutes of LLLT and sham treatments in separate upper back quadrants. The histamine model was applied to the upper quadrants, and Mucuna pruriens to the lower quadrants. Pruritus intensity, alloknesis, hyperknesis, flare area, and skin temperature were measured pre and post treatment.

### Results

Seventeen individuals (eight females, nine males) participated in the study. In the histamine model, LLLT notably reduced itch intensity (difference = 13.9 (95% CI: 10.5 − 17.4), p = 0.001), alloknesis (difference = 0.80 (95% CI: 0.58–1.02), p = 0.001), and hyperknesis (difference = 0.48 (95% CI: 0.09–0.86), p = 0.01). Skin temperature changes were not significantly different between the two groups (difference = -2.0 (95% CI: -6.7–2.6), p = 0.37). For the Mucuna pruriens model, no significant differences were observed in any measures, including itch intensity (difference = 0.8 (95% CI: -2.3 − 3.8), p = 0.61) hyperknesis

**Funding:** The author(s) received no specific funding for this work.

**Competing interests:** The authors have declared that no competing interests exist.

(difference = 0.08 (95% CI: -0.06–0.33), p = 0.16) and alloknesis (difference = 0. 0.09 (95% CI: -0.08–0.256), p = 0.27).

## Conclusions

LLLT effectively reduced histamine-induced pruritus, alloknesis, and hyperknesis; however, LLLT was ineffective against Mucuna pruriens-induced pruritus. Further investigations are required to determine LLLT's effectiveness of LLLT in various pruritus models.

## Introduction

Pruritus is a common symptom in various dermatological conditions and can significantly affect the quality of life. The annual cumulative incidence of chronic pruritus is estimated to be 7%, with a prevalence between 8% and 25.5% [1, 2]. While there are several treatment options available, their effectiveness is often limited.

Sharing broad similarities with pain, pruritus is commonly associated with somatosensory abnormalities. Itch-associated dysesthesias, such as mechanical alloknesis and hyperknesis, observed in individuals with pruritus, are indicative of neuronal sensitization processes [3, 4]. Specifically, alloknesis, analogous to pain-related allodynia, is characterized by the elicitation of itch in response to stimuli that are typically non-itch-provoking [5, 6]. In comparison, hyperknesis is an itch-related analog of the nociceptive state of hyperalgesia. Accordingly, hyperknesis describes an increased pruritic response in terms of magnitude and/or duration to a mild pruritogenic stimulus [3, 5, 7]. Although the pain analogs of these neurophysiological phenomena have gained considerable relevance in the understanding of pain mechanisms in recent decades, alloknesis and hyperknesis are still scarcely investigated [8]. Consequently, treatment of the central processing of pruritus is regarded as highly challenging [9].

Pharmacological management of pruritus traditionally relies on antihistamine therapy; nevertheless, the persistence of pruritus in conditions unresponsive to antihistaminic treatment underscores the presence of non-histaminergic pruritic pathways [10]. This recognition has led to the bifurcation of pruritus into histamine-induced and non-histamine-induced models [11]. Histamine is the most extensively studied pruritogen and is frequently used to experimental pruritus induction [11]. Regardless of the route of histamine administration, moderate to severe spontaneous pruritus is elicited, which is associated with the presence of nociceptive sensations, alloknesis, and hyperknesis [6, 12]. In addition to intense itching, vasodilation ("flare") around the application site has been described [13]. In contrast, non-histamine-dependent pruritus has been successfully induced through the application of spicules derived from African itch beans [14–16].

Photobiomodulation (PBMT), also referred as Low-level light therapy (LLLT), represents a promising non-pharmacological treatment approach. LLLT involves the application of light with wavelengths of 600–1100 nm and a power density of 5 mW/cm$^2$ to 5 W/cm$^2$ [17]. Clinical studies have shown positive effects on wound healing and tissue regeneration, scar formation, and blood flow [18–21]. In addition, LLLT is effectively reducing postoperative and chronic pain [18, 20–22]. In a human pain model, an alleviating effect of LLLT on hyperesthesia and allodynia, as well as an increase in heat pain threshold (HPT) and mechanical pain threshold (MPT) has been demonstrated. This suggests a significant modulatory effect on peripheral and central sensitization, which is attributed to a reduction in neurogenic inflammation and the associated neuronal sensitization [19]. The neurophysiological similarities of pain and pruritus

suggest that therapeutic approaches targeting central sensitization, such as the use of gabapentinoids, may also be effective in treating pruritus [23]. This raises the possibility that LLLT may also be effective in alleviating pruritus. Therefore, this study aimed to evaluate the possible influence of LLLT on both histaminergic and non-histaminergic pruritus, as well as on alloknesis and hyperknesis.

## Materials and methods

### Study design and framework

This study was designed as a prospective, randomized, sham-controlled, split-body, double-blind study in healthy volunteers. After approval of the ethics committee at the Medical University of Graz and registration of the protocol (https://classic.clinicaltrials.gov/ct2/show/NCT05369338), this study was conducted at the Medical University of Graz, Austria and was conducted in accordance with the Declaration of the World Medical Association. This article was prepared in accordance with the CONSORT Statement [24].

### Randomization and blinding

The allocation of LLLT or sham treatment was determined by a computer-generated random sequence obtained from www.randomizer.at. This randomization process assigned one side to receive LLLT and the contralateral side to receive sham treatment with an inactivate device, serving as the control group. The application of irradiation and sham treatments were administered by an individual who did not participate in any other aspect of the study.

The binding of participants was ensured through the use of opaque glasses and auditory shielding, which prevented any visual or auditory cues regarding the treatment type. The study investigators were also blinded to the treatment allocation because they did not have access to the online randomization data and were physically absent from the treatment room during the irradiation process.

Blinding of the assessor was unfeasible in the Mucuna pruriens model, in contrast to histamine, which does not induce a flare reaction. As the pruritus models were applied to the participants' backs, the participants remained blinded to which side received the irradiation treatment.

### Participants

**Inclusion criteria for the selection of the subjects.** Healthy subjects were informed of the study, and after providing written consent, they were included in the study. The study took place between April 4th to May 27th 2022.

The inclusion criteria were as follows:

- Able to provide written informed consent

- Healthy male and female individuals between the ages of 18 and 60

- Female participants with a negative pregnancy test

  The exclusion criteria were as follows:

- known allergy or hypersensitivity to histamine or the African scratch bean

- anamnestic presence of skin diseases

- piercings

- fever

- tattoos in the test area

- neoplasia in the test area

- pregnancy

- pacemaker

- local acute infections, skin inflammations, skin lesions

- epilepsy

## Pruritus paradigms

Two pruritus models were applied to each participant. The upper back of each patient was divided into four quadrants. The histamine model was applied to the two upper quadrants and the Mucuna pruriens model was applied to the two lower areas. Immediately after the application of the first pruritus model, a baseline measurement was performed. The individual responsible for randomization positioned the LLLT device (Repuls7, Repuls Lichtmedizintechnik GmbH, Vienna, Austria) simultaneously with a 7 cm ring in direct contact with the skin. According to the randomization plan, the LLLT device was activated or deactivated as a sham treatment.

Each patient underwent four experiments in a randomized order: histamine model with LLLT and sham treatment and M. pruriens with LLT and sham treatment. Experiments were followed by a one hour break to avoid a carry-over effect. The course of this study is shown in Fig 1.

**Histamine application.** We used the histamine model described by Darsow et al. [25] After application of a drop of histamine gel (1% histamine dihydrochloride in 2.5% methylcellulose), the skin was superficially punctured with a conventional blood lancet [25]. Application was always performed by the same staff member to minimize the variability in the application technique.

**Mucuna pruriens application.** The spicules of Mucuna pruriens (African scratch bean) are a proven model for eliciting histamine-independent pruritus. We used a methodology similar to the model described by Papoiu et al. [26] An area of 2x2 cm was taped off with bandages to prevent spicules from spreading to the surrounding skin. In this skin area, 40 to 45 spicules, previously counted under the microscope, were applied with micro tweezers and circularly rubbed onto the skin for 45 seconds [26].

## Interventions

Subjects were exposed to the application of LLLT for 6 minutes (histamine-induced and non-histamine-induced pruritus area) based on the manufacturer's recommendations. Repuls 7 (Repuls Lichtmedizintechnik GmbH, Vienna, Austria) is a class IIb medical device that applies pulsed red light at a frequency of 640 nm. The intensity was 175 mW cm$^{-2}$, corresponding to a power density of 4100 mW. The pulse frequency was set at 2.5 Hz. The device was placed 7 cm above skin level using an opaque spacer ring, which also prevented accidental irradiation of surrounding tissue. In the sham treatment, the device was also placed over the itchy area using a spacer ring but remained inactive.

## Measurements

The assessment was performed immediately after LLLT application was completed. The sequence of tests corresponded to the order in which they were presented. All participants

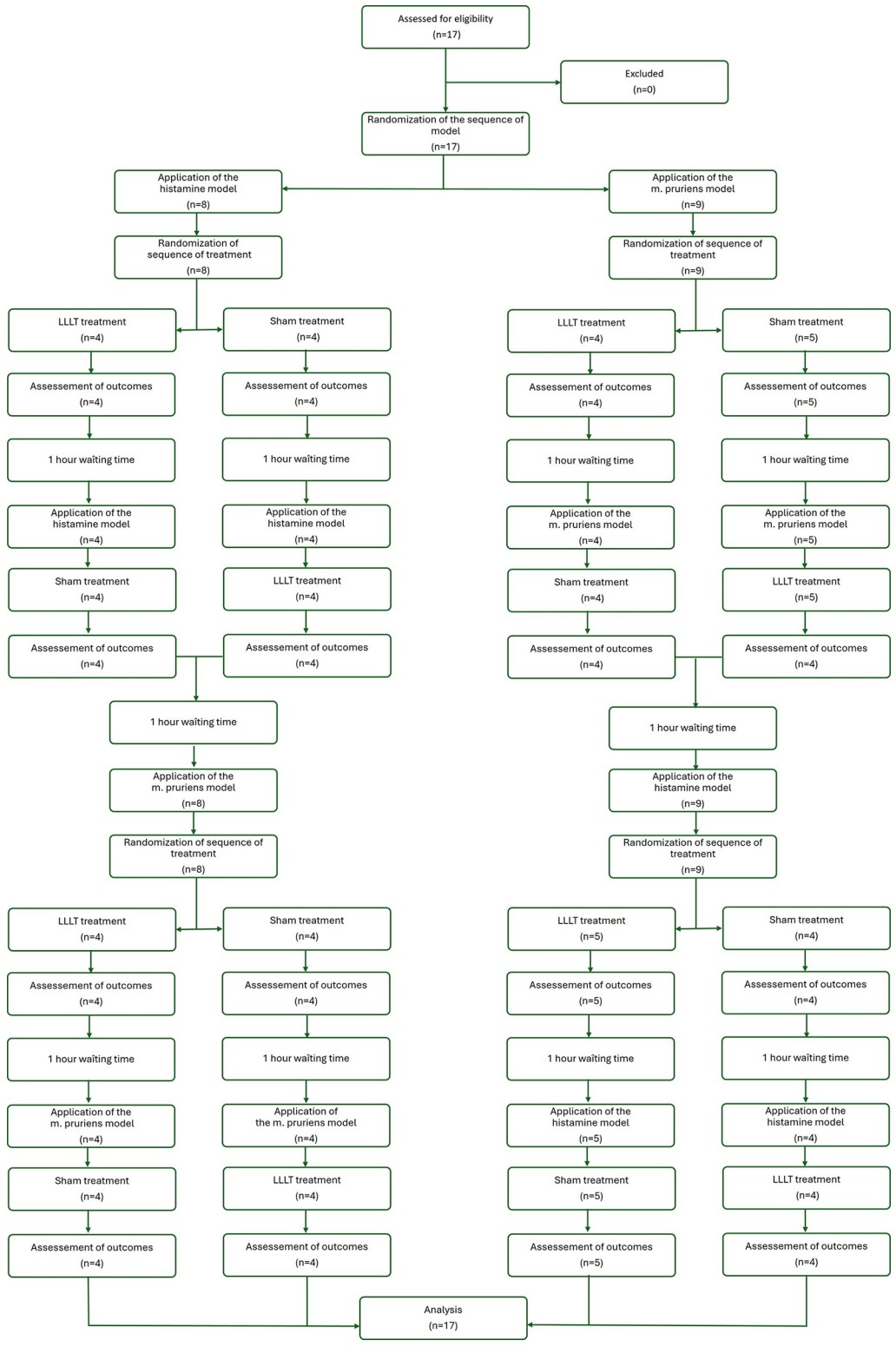

**Fig 1. Flowchart of the study.**

were instructed and trained on the modalities before the first examination to ensure that the examination could be conducted as promptly as possible.

The examination included pruritus intensity as a primary parameter. The distance of alloknesis and hyperkensia, flare area, and skin temperature were secondary outcomes.

**Intensity of pruritus.** The intensity of pruritus was measured using a 101-point visual analog scale (VAS 0–100). Participants were instructed to rate the intensity of maximum pruritus during the experiment.

**Alloknesis.** The alloknesis distance was measured using a cotton swab to repeatedly touch the skin along four radial lines in a series of short strokes. Swabbing began 10 cm from the injection site/center of pruritic substance application and continued in the direction of the injection site until the subject reported a sensation of itching or the strokes came within 1 cm of the injection site [27]. The measurements were averaged and reported as mean distance.

**Hyperknesis.** Hypeknesis was assessed using a von Frey filament. This was placed 10 cm from the centre of application of the pruritogenic substance centrally along four radial lines spaced 0.5 cm apart. Subjects were instructed to ignore the initial stinging sensation and rate only the severity of itching that occurred on a visual analog scale. The measurements were averaged and reported as mean distance [27].

**Flare area.** The size of the flare area was determined photographically and thermographically using a thermal imaging system (Flir E5 Pro,Flir Systems Inc., Wilsonville, Oregon, USA). The size of the flare area was calculated using the ImageJ software (National Institutes of Health, Bethesda, USA; ImageJ is in the public domain and can be downloaded from the ImageJ website (http://rsb.info.nih.gov/ij). As Mucuna pruriens was repeatedly shown not to induce a flare reaction, no thermography was performed [28].

**Skin temperature.** The skin temperature was measured at the center of the flare area using a thermographic device.

**Adverse effects.** The participants were asked about potential side effects at each study visit.

## Monitoring

To ensure quality control, independent clinical monitors from the "Koordinierungszentrum für klinische Studien" of the Medical University of Graz (KKS, Clinical Trials Unit) conducted on-site data monitoring during the study. This monitoring aimed to ensure patient safety, adherence to protocols, Good Clinical Practice (GCP), and consistency of the data.

## Statistics

**Sample size calculation.** The sample size was calculated based on the following assumptions: In the publication by Arendt-Nielsen et al., pruritus severity was reported as 32.9 ± 5.6. on a 101-point VAS scale. A reduction of more than 15% was defined as clinically relevant. This resulted in a minimum sample size of 15 participants for a two-sided paired t-test. The significance level alpha was defined as 0.05, and beta as 0.1. The sample size was calculated using G*Power 3.1.9.5 [29]. To account for 10% potential dropout, the sample size was increased to 17 subjects.

## Statistical analysis

A two-tailed t-test for dependent samples was used to evaluate the mean differences between the paired groups. Statistical significance was set at a level of $p < 0.05$.

**Table 1. Demographics of the study population.**

|  | Mean (min-max) or n/N (%) |
|---|---|
| **Age (years)** | 32.5 (19–61) |
| **Height (cm)** | 175.8 (161–195) |
| **Weight (kg)** | 71.8 (50–105) |
| **Sex, Women** | 8 (47.1) |
| **Fair skin Caucasian** | 17 (100) |
| **Fitzpatrick classification Type IV** | 17 (100) |

N, number; SD, standard deviation

No side effects or adverse events were reported in any treatment group.

## Results

Seventeen healthy volunteers (eight female and nine male) were screened for eligibility and enrolled in the study (see Fig 1). The demographic data of the patients are presented in Table 1. The ages of the subjects ranged from 19 to 65 years. All participants were Caucasians with fair skin, which corresponds to skin type IV of the Fitzpatrick classification [30]. All the participants completed the study without deviating from the protocol.

Table 2 presents the comparative results of pruritus assessments between LLLT and sham treatments across different metric. In the histamine model, significant reductions were observed with LLLT in itch intensity, spatial area of itch and enhanced sensation of itch. However, temperature changes between LLLT and sham were not significant. A significant difference was also noted in the area of flare. In contrast, the Mucuna pruriens model showed no significant differences between LLLT and sham treatments in itch intensity, enhanced sensation of itch, and spatial area of itch. These results suggest that LLLT was significantly effective in reducing histamine-induced pruritus compared with sham treatment, whereas its effects were less pronounced and not statistically significant in the Mucuna pruriens model.

## Discussion

Based on our findings, LLLT has a significant effect on histamine-induced pruritus. In contrast, LLLT did not show any effect on Mucuna pruriens-induced pruritus.

Recent meta-analytic evidence attests to the clinical effectiveness of LLLT for various indications, including acute radiation dermatitis [31] and diabetic foot ulcers [32]. Nonetheless,

**Table 2. Results of pruritus workup, presented as mean value ± standard deviation.** The values correspond to the difference between the measurements before and after the low-level light and the sham treatment in the histamine and the Mucuna pruriens model.

|  | LLLT Mean | Sham Mean | Mean Difference | 95% CI of the Difference | p-Value |
|---|---|---|---|---|---|
| **Histamine Itch VAS (0–100)** | 30.5 | 16.6 | 13.9 | 10.5 – 17.4 | 0.001 |
| **Histamine Alloknesia (cm)** | 0.80 | 0.06 | 0.80 | 0.58–1.02 | 0.001 |
| **Histamine Hyperknesia (cm)** | 0.60 | 0.12 | 0.48 | 0.09–0.86 | 0.01 |
| **Histamine Temperature (˚C)** | 2.3 | 0.2 | -2.0 | -6.7–2.6 | 0.37 |
| **Histamine Flare (cm²)** | 0.35 | 0.17 | 0.18 | 0.09–0.27 | 0.001 |
| **M. pruriens VAS (0–100)** | 7.5 | 6.8 | 0.8 | -2.3 – 3.8 | 0.61 |
| **M. pruriens Alloknesia (cm)** | 0.17 | 0.09 | 0.09 | -0.08–0.256 | 0.27 |
| **M. pruriens Hyperknesia (cm)** | 0.23 | 0.08 | 0.08 | -0.06–0.33 | 0.16 |

CI, Convidence Interval; LLLT, Low-Level Light Therapy; VAS, Visual Analogue Scale

despite half a century of research into photobiomodulation, the precise mechanisms by which LLLT achieves its therapeutic effects remain a subject of debate [17].

Several studies have discussed the influence of LLLT on histamine release, which could offer a possible explanation for its pruritus-reducing effect. In animal experiments, LLLT was shown to significantly affect histamine-induced edema compared with the control group, consistent with the repeatedly demonstrated photosensitivity of mast cells [33, 34]. The effect of the LLLT on the state of mast cell activation may also be of importance [33, 35, 36]. An increase in activated mast cells was shown in inflammatory and proliferative phases, while mast cell numbers in the remodeling phase were reduced by the application of LLLT [37].

Furthermore, LLLT influences inflammatory pathways e.g. by inhibiting cyclooxygenase-2, interleukin-6, prostaglandin E2, tumor necrosis factor α, and matrix metalloproteinase 1 and 3 [38]. This represents and additional relevant histamine-independent mechanism by which LLLT may reduce pruritus.

Based on these considerations, LLLT has already been clinically used for the treatment of pruritus. The evidence, however, is sparse. In a case series, Olivera et al. showed the effectiveness of LLLT in alleviating pruritus among burn victims [39]. In a prospective observational study of patients with atopic dermatitis, LLLT led to a reduction of symptoms, including itching [40]. In a recent small randomized clinical trial in patients with renal pruritus, the combination of LLLT plus antihistamine medication was compared to antihistamines alone [41]. This study is of particular interest, as based on their data, it can be assumed that LLLT has an effect beyond the histamine pathway. In our study, we aimed to elucidate this issue by including a histamine-independent pruritus model. However, LLLT failed to affect Mucuna pruriens induced pruritus. Besides the obvious explanation that LLLT cannot influence histamine-independent itching, another reason could be that in our study the symptoms induced with Mucuna pruriens were only very mild. Therefore, the effect may not have been statistically significant. Previous studies have shown inconsistent results regarding the intensity of itch sensation induced by Mucuna pruriens. In some instances, it has been reported to cause a stronger itch than histamine [26], while in others, histamine has had the more potent effect [4]. One potential factor could be the variability in the quality of the herbal component used in these studies.

The influence of LLLT on the flare area and temperature was present to a significant extent in the histamine group compared with the control group. Mucuna pruriens induced pruritus is mediated primarily by a subpopulation of mechano-heat-sensitive/polymodal C-fibers, which do not produce a flare area or temperature change. Therefore, we cannot report any data in this regard.

In the field of pain medicine, allodynia and mechanically induced hyperalgesia are regarded as markers of central sensitization processes, triggered by an amplification cascade. This typically results in changes of mechanical sensitivity parameters [42, 43]. LLLT had previously been shown to be capable of influencing this process. [19]. Given the neurophysiological similarities between pain and pruritus [3], it is hypothesized that LLLT might also affect the central mechanisms underlying pruritus. This is based on commonalities in the neural pathways involved in the processing of pain and itch stimuli, which can lead to an exaggerated sensory response. Conditions such as pruritus, akin to pain, may emanate from alterations at both the spinal and supraspinal levels, evident in symptoms such as hyperknesis and alloknesis [44].

Our research on the effects of LLLT on these pruritic manifestations, particularly in the context of a histamine-induced model, revealed a significant decrease in both hyperknesis and alloknesis. This supports the hypothesis that LLLT's therapeutic actions may extend to modulating the central neural mechanisms involved in itch, akin to those in pain, thereby offering a novel approach to managing pruritus. These findings not only reinforce the potential of LLLT in pain management, but also open new avenues for its application in treating pruritus.

## Limitations

The results of our study must be interpreted within the context of the selected study population. The study design was structured to minimize confounding variables by selecting healthy individuals without any purities-related medications or conditions, to isolate the underlying mechanisms more clearly. However, this approach does not mirror the typical clinical demographic in need of therapeutic intervention.

A notable limitation of our results was the modest effect observed with the Mucuna pruriens model compared to histamine. Previous literature suggests that the pruritogenic effects of Mucuna pruriens and histamine show large variability among individuals. Given that Mucuna pruriens is derived from a plant, as such its effects are inherently variable [45]. In our study, the observed induced effects were minimal, resulting in inconclusive treatment outcomes. This underscores the need for a cautious interpretation of these findings. Additionally, the effects on histamine-independent pruritus require further investigation using a alternative research model.

An other limitation is that we have not assessed to development of pruritus over time. We therefore cannot exclude the possibility that part of the effect is due to the fact that the histamine and Mucuna pruriens weakened in their pruritus induction over time [3]. However, this effect should affect both the Sham and the LLLT group.

The wash-out period between the study arms was chosen to be 1 hour. This is a reasonable time frame in regards to the pruritus paradigms. For the photobiomodulation however, a longer interval would be required to guarantee that no residual effect is present. However, by randomizing the order of both, the pruritus model and the treatment, we have at least minimized effect. Even more so, we switched the area of treatment and used an opaque filter to prevent scattered irradiation of the surrounding tissue. However, we still can not completely exluce an unwanted residual systemic effect between the experiments.

Furthermore, it is crucial to consider the specific set of parameters of the LLLT device we used. The Repuls 7 exhibits an irradiance level of 175mW/cm$^2$, which has the potential to induce thermal effects in tissues. This level of irradiance exceeds the typical threshold for photobiomodulation, which is defined as not surpassing 100mW/cm$^2$ in the skin. Likewise, the pulse frequence of 2.5Hz may also have a specific influence. Therefore, the observed effects should not be generalized uncritically to other devices or setups.

## Clinical application and future perspective

The transferability of our findings from short-lasting models to clinical practice in the treatment of chronic pruritus is undoubtedly restricted. Nevertheless, our study sheds light on the pathophysiology of a non-pharmacological treatment modality that has been previously evaluated in clinical settings [39–41]. The findings from our study encourages the clinical utilisation of LLLT in the treatment of pruritus, specifically when triggered by histamine. It has however to been prone in mind that our study evaluated the effect of a single session of LLLT treatment. We can only speculate on the duration of the effect, or of the course of repeated treatments.

## Conclusion

The present study demonstrates that LLLT has clinically relevant antipruritic properties in histamine-induced pruritus. It has been shown to alleviate alloknesis and hyperknesis. We failed to observe any effect on Mucuna pruriens-induced pruritus.

## Supporting information

**S1 Checklist. CONSORT 2010 checklist of information to include when reporting a randomised trial\*.**
(DOC)

**S1 File.**
(PDF)

**S2 File.**
(PDF)

## Author Contributions

**Conceptualization:** Kordula Lang-Illievich, Helmar Bornemann-Cimenti.

**Formal analysis:** Heike Schulze-Bauer.

**Investigation:** Kordula Lang-Illievich, Heike Schulze-Bauer.

**Methodology:** Kordula Lang-Illievich, Christoph Klivinyi, Helmar Bornemann-Cimenti.

**Project administration:** Helmar Bornemann-Cimenti.

**Validation:** Christoph Klivinyi.

**Visualization:** Ala Elhelali.

**Writing – original draft:** Kordula Lang-Illievich, Christoph Klivinyi, Helmar Bornemann-Cimenti.

**Writing – review & editing:** Heike Schulze-Bauer, Ala Elhelali.

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
