## [Decision Letter · Decision Letter 0]

8 May 2024

PONE-D-24-12900The effect of low-level light therapy on histamine and Mucuna pruriens-induced pruritus, hyperknesis and alloknesis in healthy volunteers: A double-blind, randomized, sham-controlled studyPLOS ONE

Dear Dr. Bornemann-Cimenti,

Thank you for submitting your manuscript to PLOS ONE. After careful consideration, we feel that it has merit but does not fully meet PLOS ONE’s publication criteria as it currently stands. Therefore, we invite you to submit a revised version of the manuscript that addresses the points raised during the review process.

All three reviewers have each made criticisms that must be thoroughly addressed in any resubmission. One reviewer recommended rejection so this is still a possibility Please submit your revised manuscript by Jun 22 2024 11:59PM. If you will need more time than this to complete your revisions, please reply to this message or contact the journal office at plosone@plos.org. Please include the following items when submitting your revised manuscript:A rebuttal letter that responds to each point raised by the academic editor and reviewer(s). You should upload this letter as a separate file labeled 'Response to Reviewers'.A marked-up copy of your manuscript that highlights changes made to the original version. You should upload this as a separate file labeled 'Revised Manuscript with Track Changes'.An unmarked version of your revised paper without tracked changes. You should upload this as a separate file labeled 'Manuscript'.

We look forward to receiving your revised manuscript.

Kind regards,

Michael R Hamblin

Academic Editor

PLOS ONE

Journal Requirements:

Reviewers' comments:

Reviewer's Responses to Questions

**Comments to the Author**

1. Is the manuscript technically sound, and do the data support the conclusions?

Reviewer #1: Partly

Reviewer #2: Yes

Reviewer #3: Partly

2. Has the statistical analysis been performed appropriately and rigorously? 

Reviewer #1: No

Reviewer #2: Yes

Reviewer #3: Yes

3. Have the authors made all data underlying the findings in their manuscript fully available?

Reviewer #1: Yes

Reviewer #2: Yes

Reviewer #3: No

4. Is the manuscript presented in an intelligible fashion and written in standard English?

Reviewer #1: Yes

Reviewer #2: Yes

Reviewer #3: Yes

5. Review Comments to the Author

Reviewer #1: Major Revision

This is a very interesting investigation of the potential use of Low-Level Light Therapy (LLLT) in patients with a form of pruritis. However, the analysis, as reported, requires revision particularly the calculation of confidence intervals for the outcome measurers are required. Some other points that need attention are listed:

Lines

30-33 It would have been useful to include the duration of the LLLT/Sham and also the fact that the outcome measures taken were assessed pre- and post- treatment.

35-42 This will need a total rewrite taking account of the points indicated below

210 Need to include the units for the scale used for the measure of 32.9

216-220 In a Pre and Post treatment study with paired observations made on each subject, it is not important to check the Pre and Post data individually for the Normal distributions of each. For each patient, it is the change d = Post − Pre which is of interest. Whatever the distribution form of Pre and Post, d will tend to have a Normal distribution. [This is also true if the data are non-parametric.] Thus, the tests discussed here are not relevant.

What is now important is to determine the mean of these differences, ¯d and then do the paired t-test of the null hypothesis of 0 difference to get the p-value.

More essential however, is to calculate and report the corresponding 95% Confidence Interval (CI).

226 According to Table 1 this should be Type V

230 Table 1: For describing characteristics of the patients, it is better to quote (say) the mean, age with youngest and oldest patient ages rather than SD. Similarly for height, and weight

230 Table 1: Would it be better to replace ‘Race, White’ by ‘Fair skin Caucasian’?

230 Table 1: Suggest omit: Types I, II, III, IV and VI

233-247 This will need a total rewrite taking account of some of the points raised

251-252 Table 2: The difference d = LLLT – Sham, needs presenting with its 95% CI, and the p-value. For example, Histamine VAS: d = 30.5 − 16.6 =13.9 with 95%CI XX to YY, p = 0.00001.[I cannot calculate the CI or p-value without the 13 paired differences.]

251-252 Table 2: ‘Itch’ should be added to the second row

251-252 Table 2 and elsewhere: P-values should be quoted to two significant places. So, for example, ‘0.75774 and 0.07960’ become ‘0.76 and 0.080’ respectively

Reviewer #2: This is a very elegantly designed and clearly presented work on the efficacy of LLLT on two different types of pruritus: histamine-induced and non-histamine-induced, in a split-body and sham-controlled model. The intensity of pruritus, and the spacial extent of two itch sensation phenomenon, including alloknesis and hyperknessis were evaluated in this trial, which is potentially of interest to the management of chronic itch. I would like to suggest a revised manuscript before accepted for publication. My concerns and comments are as follows:

Materials and methods sections

1.As for LLLT intervention, is there any shelter or other methods to eliminate sparing effect between quadrants?

2.Previous studies has demonstrated the acute itch evoked by whether histamine or cowhage only lasts for 5-20 minutes. How long the induced-itch last in this trial? And please clarify whether the itch sensation subside itself or reduced by LLLT?

Andersen HH et al. Alloknesis and hyperknesis-mechanisms, assessment methodology, and clinical implications of itch sensitization. Pain. 2018;159(7):1185-1197.

3.The calculation of simple size has been presented in supporting information. I recommend that it should be included in the manuscript.

Discussion section

1.The 3rd paragraph presented interpretation of the effect of LLLT in reducing itch, however, only its influence on mast cell was discussed. I suggest authors should provide with more evidence on more key factors in pruritus pathway, including inflammatory and nonhistamine factors.

2. There is only 1 session of LLLT administered to subjects. I recommend that the extent of clinical significance of these result should be discussed.

Typos

1.Line 27. It should be “hyperknesis”, rather than “hyperkinesis”. There are some of the same typo in this manuscript which should be revised.

2.Line 70. It seems that the reference number “(11)” is large in font size. And some punctuation surrounding the reference number in the main text were not correct.

3.Table 2. There should be a space between “M. pruritus” and “Alloknesia”.

4.Line 308. The full comma was missing at the end of this paragraph.

Reviewer #3: Your paper is well-crafted and engaging. However, I have several concerns that influence my decision not to endorse your study for publication. Firstly, the terminology used for Low-Level Light Therapy (LLLT) has been updated to photobiomodulation. Secondly, the LED device you employed (Repuls 7) exhibits an irradiance level of 175mW/cm², potentially inducing thermal effects in tissues. This level of irradiance exceeds the typical threshold for photobiomodulation, defined as not surpassing 100mW/cm² in the skin. Furthermore, you did not provide measurements of actual irradiance using a power meter from a distance of 7cm from the skin surface. The relationship between irradiance and distance is crucial, following an inverse square law, meaning irradiance decreases as distance from the light source increases. Assuming a power of 4100W at 1cm, the actual irradiance would approximate 83mW/cm², resulting in a fluence of 30J/cm² over 6 minutes—a detail not mentioned in your study. Additionally, your choice to use a pulsing frequency of 2.5Hz raises questions, as a continuous wave (CW) might have been more appropriate to avoid introducing an unverified variable. Beam uniformity, including potential hot and cold spots, was also not addressed. Lastly, considering that photobiomodulation effects can be both local and systemic, a washout period of only one hour between procedures seems inadequate. The effects of photobiomodulation are typically observed between 24 to 48 hours post-exposure, suggesting a need for a longer interval.

6. PLOS authors have the option to publish the peer review history of their article (what does this mean?). If published, this will include your full peer review and any attached files.

Reviewer #1: No

Reviewer #2: No

Reviewer #3: No

---

## [Author Response · Author response to Decision Letter 0]

21 May 2024

Review Comments to the Author

Reviewer #1:

The analysis, as reported, requires revision particularly the calculation of confidence intervals for the outcome measurers are required. 

We have now included the CO intervals. 

Lines

30-33 It would have been useful to include the duration of the LLLT/Sham and also the fact that the outcome measures taken were assessed pre- and post- treatment.

We have added these information. 

35-42 This will need a total rewrite taking account of the points indicated below

We have rewritten this section.

210 Need to include the units for the scale used for the measure of 32.9

We added this information.

216-220 In a Pre and Post treatment study with paired observations made on each subject, it is not important to check the Pre and Post data individually for the Normal distributions of each. For each patient, it is the change d = Post − Pre which is of interest. Whatever the distribution form of Pre and Post, d will tend to have a Normal distribution. [This is also true if the data are non-parametric.] Thus, the tests discussed here are not relevant.

We have erased this passage and re-calculated the data accordingly. 

What is now important is to determine the mean of these differences, ¯d and then do the paired t-test of the null hypothesis of 0 difference to get the p-value.

More essential however, is to calculate and report the corresponding 95% Confidence Interval (CI).

We are now reporting the results according to the paired-t-test including the 95%-CI of the difference.

226 According to Table 1 this should be Type VI

We have corrected this issue.

230 Table 1: For describing characteristics of the patients, it is better to quote (say) the mean, age with youngest and oldest patient ages rather than SD. Similarly for height, and weight

We have now included minimum and maximum values for these measures. 

230 Table 1: Would it be better to replace ‘Race, White’ by ‘Fair skin Caucasian’?

We have changed the wording

230 Table 1: Suggest omit: Types I, II, III, IV and VI

We have omitted the unnecessary types. 

233-247 This will need a total rewrite taking account of some of the points raised

We have rewritten this paragraph.

251-252 Table 2: The difference d = LLLT – Sham, needs presenting with its 95% CI, and the p-value. For example, Histamine VAS: d = 30.5 − 16.6 =13.9 with 95%CI XX to YY, p = 0.00001.[I cannot calculate the CI or p-value without the 13 paired differences.]

We have now reported the results according to these suggestions.

251-252 Table 2: ‘Itch’ should be added to the second row

We added the word itch. 

251-252 Table 2 and elsewhere: P-values should be quoted to two significant places. So, for example, ‘0.75774 and 0.07960’ become ‘0.76 and 0.080’ respectively

We have changed the reporting or the p-values throughout the manuscript.

Reviewer #2: 

Materials and methods sections

1.As for LLLT intervention, is there any shelter or other methods to eliminate sparing effect between quadrants?

The spacer ring was opaque and prevented accidental irradiation of surrounding tissue. We added this information to the methodology. 

2.Previous studies has demonstrated the acute itch evoked by whether histamine or cowhage only lasts for 5-20 minutes. How long the induced-itch last in this trial? And please clarify whether the itch sensation subside itself or reduced by LLLT? [Andersen HH et al. Alloknesis and hyperknesis-mechanisms, assessment methodology, and clinical implications of itch sensitization. Pain. 2018;159(7):1185-1197.]

As we have not assessed these data overtime, we cannot provide it. We have however added this point in the limitation section. 

3.The calculation of simple size has been presented in supporting information. I recommend that it should be included in the manuscript.

The sample size calculation is now in the statistics section.

Discussion section

1.The 3rd paragraph presented interpretation of the effect of LLLT in reducing itch, however, only its influence on mast cell was discussed. I suggest authors should provide with more evidence on more key factors in pruritus pathway, including inflammatory and nonhistamine factors.

We now have broadened this section and included considerations on non-histamine factors.

2. There is only 1 session of LLLT administered to subjects. I recommend that the extent of clinical significance of these result should be discussed.

We added this point in the “future perspective section”.

Typos

1.Line 27. It should be “hyperknesis”, rather than “hyperkinesis”. There are some of the same typo in this manuscript which should be revised.

We have corrected this issue throughout the manuscript.

2.Line 70. It seems that the reference number “(11)” is large in font size. And some punctuation surrounding the reference number in the main text were not correct.

We have corrected this issue.

3.Table 2. There should be a space between “M. pruritus” and “Alloknesia”.

We have corrected this issue.

4.Line 308. The full comma was missing at the end of this paragraph.

We have corrected this issue.

Reviewer #3: 

Firstly, the terminology used for Low-Level Light Therapy (LLLT) has been updated to photobiomodulation.

We have now added the exprecession in the title, abstract and introduction.

 Secondly, the LED device you employed (Repuls 7) exhibits an irradiance level of 175mW/cm², potentially inducing thermal effects in tissues. This level of irradiance exceeds the typical threshold for photobiomodulation, defined as not surpassing 100mW/cm² in the skin. Furthermore, you did not provide measurements of actual irradiance using a power meter from a distance of 7cm from the skin surface. The relationship between irradiance and distance is crucial, following an inverse square law, meaning irradiance decreases as distance from the light source increases. Assuming a power of 4100W at 1cm, the actual irradiance would approximate 83mW/cm², resulting in a fluence of 30J/cm² over 6 minutes—a detail not mentioned in your study. Additionally, your choice to use a pulsing frequency of 2.5Hz raises questions, as a continuous wave (CW) might have been more appropriate to avoid introducing an unverified variable. Beam uniformity, including potential hot and cold spots, was also not addressed. 

We concur with the reviewer's assertion that it would have been intriguing to ascertain the precise power output. However, we are unable to provide the requisite data. Nevertheless, we have incorporated this aspect into the limitations section. 

Lastly, considering that photobiomodulation effects can be both local and systemic, a washout period of only one hour between procedures seems inadequate. The effects of photobiomodulation are typically observed between 24 to 48 hours post-exposure, suggesting a need for a longer interval.

By using a randomization schema for both, the pruritus model and the treatment order and by switching between four different treatment sides we hoped to minimize this issue. Nevertheless, we agree with the reviewer that there is a risk of a systemic carry-over effect. We have included this in our limitations accordingly.

---

## [Decision Letter · Decision Letter 1]

19 Jun 2024

PONE-D-24-12900R1The effect of photobiomodulation on histamine and Mucuna pruriens-induced pruritus, hyperknesis and alloknesis in healthy volunteers: A double-blind, randomized, sham-controlled studyPLOS ONE

Dear Dr. Bornemann-Cimenti,

Thank you for submitting your manuscript to PLOS ONE. After careful consideration, we feel that it has merit but does not fully meet PLOS ONE’s publication criteria as it currently stands. Therefore, we invite you to submit a revised version of the manuscript that addresses the points raised during the review process.

Reviewer 1 has provided guidance on the changes to be made. reviewer 2 requests a figure to be added

We look forward to receiving your revised manuscript.

Kind regards,

Michael R Hamblin

Academic Editor

PLOS ONE

Reviewers' comments:

Reviewer's Responses to Questions

**Comments to the Author**

1. If the authors have adequately addressed your comments raised in a previous round of review and you feel that this manuscript is now acceptable for publication, you may indicate that here to bypass the “Comments to the Author” section, enter your conflict of interest statement in the “Confidential to Editor” section, and submit your "Accept" recommendation.

Reviewer #1: (No Response)

Reviewer #2: All comments have been addressed

2. Is the manuscript technically sound, and do the data support the conclusions?

Reviewer #1: Yes

Reviewer #2: Yes

3. Has the statistical analysis been performed appropriately and rigorously? 

Reviewer #1: No

Reviewer #2: Yes

4. Have the authors made all data underlying the findings in their manuscript fully available?

Reviewer #1: Yes

Reviewer #2: Yes

5. Is the manuscript presented in an intelligible fashion and written in standard English?

Reviewer #1: Yes

Reviewer #2: Yes

6. Review Comments to the Author

Reviewer #1: Major Revision but one that can be easily conducted

In the earlier version of this paper the Abstract quantified some of the differences between the LLLT and Sham groups as follows.

“Seventeen individuals (eight females, nine males) participated in the study. In the histamine model, LLLT notably reduced itch intensity (VAS: 30.5 ± 15.0 for LLLT vs. 16.6 ± 11.7 for sham, p=0.00001), alloknesis (0.86 ± 0.36 cm for LLLT vs. 0.08 ± 0.13 cm for sham, p=0.00027), and hyperknesis (0.60 ± 0.50 cm for LLLT vs. 0.06 ± 0.30 cm for sham, p=0.00006). Skin temperature changes were not significantly different between the two groups (p=0.36731). For the Mucuna pruriens model, no significant differences were observed in any measures, including itch intensity (VAS: 7.5 ± 10.6 for LLLTvs. 6.8 ± 7.0 for sham, p=0.60616) and alloknesis (0.21 ± 0.23 cm for LLLT vs. 0.19 ± 0.16 cm for sham, p=0.75774).”

This is replaced in this revision by the following which gives no quantification to the differences observed. This is totally unsatisfactory.

“Seventeen individuals (eight females, nine males) participated in the study. In the histamine model, LLLT notably reduced itch intensity, alloknesis, and hyperknesis. Skin temperature changes were not significantly different between the two groups. For the Mucuna pruriens model, no significant differences were observed in any measures, including itch intensity and alloknesis.”

What is needed, is a simple rewrite of the first version of the Abstract. For example, replacing VAS: 30.5 ± 15.0 for LLLT vs. 16.6 ± 11.7 for sham, p=0.00001), by the magnitude of the difference between them (here 13.9) with the corresponding 95% CI.

Reviewer #2: The authors have addressed the reviewers' questions and have made appropriate revisions to the manuscript. I recommend accepting this manuscript for publication. However, there're two suggestion to improve the manuscript:

1. The manuscript could be enhanced by including an illustrative figure of the induced pruritus models.

2. In the discussion section (Lines 260-274), the authors primarily discuss diseases that are histamine-independent. It would be better to also discuss histamine-dependent pruritus, such as urticaria, if there is relevant studies.

7. PLOS authors have the option to publish the peer review history of their article (what does this mean?). If published, this will include your full peer review and any attached files.

Reviewer #1: No

Reviewer #2: No

---

## [Author Response · Author response to Decision Letter 1]

24 Jun 2024

Reviewer #1:

Reviewer #1: Major Revision but one that can be easily conducted

In the earlier version of this paper the Abstract quantified some of the differences between the LLLT and Sham groups as follows.

“Seventeen individuals (eight females, nine males) participated in the study. In the histamine model, LLLT notably reduced itch intensity (VAS: 30.5 ± 15.0 for LLLT vs. 16.6 ± 11.7 for sham, p=0.00001), alloknesis (0.86 ± 0.36 cm for LLLT vs. 0.08 ± 0.13 cm for sham, p=0.00027), and hyperknesis (0.60 ± 0.50 cm for LLLT vs. 0.06 ± 0.30 cm for sham, p=0.00006). Skin temperature changes were not significantly different between the two groups (p=0.36731). For the Mucuna pruriens model, no significant differences were observed in any measures, including itch intensity (VAS: 7.5 ± 10.6 for LLLTvs. 6.8 ± 7.0 for sham, p=0.60616) and alloknesis (0.21 ± 0.23 cm for LLLT vs. 0.19 ± 0.16 cm for sham, p=0.75774).”

This is replaced in this revision by the following which gives no quantification to the differences observed. This is totally unsatisfactory.

“Seventeen individuals (eight females, nine males) participated in the study. In the histamine model, LLLT notably reduced itch intensity, alloknesis, and hyperknesis. Skin temperature changes were not significantly different between the two groups. For the Mucuna pruriens model, no significant differences were observed in any measures, including itch intensity and alloknesis.”

What is needed, is a simple rewrite of the first version of the Abstract. For example, replacing VAS: 30.5 ± 15.0 for LLLT vs. 16.6 ± 11.7 for sham, p=0.00001), by the magnitude of the difference between them (here 13.9) with the corresponding 95% CI.

We have now rewritten the abstract section accordingly.

Reviewer #2: 

The authors have addressed the reviewers' questions and have made appropriate revisions to the manuscript. I recommend accepting this manuscript for publication. However, there're two suggestion to improve the manuscript:

1. The manuscript could be enhanced by including an illustrative figure of the induced pruritus models.

Unfortunately, we do not have any photos or illustration from the model.

---

## [Decision Letter · Decision Letter 2]

28 Jun 2024

The effect of photobiomodulation on histamine and Mucuna pruriens-induced pruritus, hyperknesis and alloknesis in healthy volunteers: A double-blind, randomized, sham-controlled study

PONE-D-24-12900R2

Dear Dr. Bornemann-Cimenti,

We’re pleased to inform you that your manuscript has been judged scientifically suitable for publication and will be formally accepted for publication once it meets all outstanding technical requirements.

Kind regards,

Michael R Hamblin

Academic Editor

PLOS ONE

Additional Editor Comments (optional):

Reviewers' comments:

Reviewer's Responses to Questions

**Comments to the Author**

1. If the authors have adequately addressed your comments raised in a previous round of review and you feel that this manuscript is now acceptable for publication, you may indicate that here to bypass the “Comments to the Author” section, enter your conflict of interest statement in the “Confidential to Editor” section, and submit your "Accept" recommendation.

Reviewer #1: All comments have been addressed

2. Is the manuscript technically sound, and do the data support the conclusions?

Reviewer #1: Yes

3. Has the statistical analysis been performed appropriately and rigorously? 

Reviewer #1: Yes

4. Have the authors made all data underlying the findings in their manuscript fully available?

Reviewer #1: Yes

5. Is the manuscript presented in an intelligible fashion and written in standard English?

Reviewer #1: Yes

6. Review Comments to the Author

Reviewer #1: (No Response)

7. PLOS authors have the option to publish the peer review history of their article (what does this mean?). If published, this will include your full peer review and any attached files.

Reviewer #1: No

---

## [Editor Report · Acceptance letter]

9 Jul 2024

PONE-D-24-12900R2 

PLOS ONE

Dear Dr. Bornemann-Cimenti, 

I'm pleased to inform you that your manuscript has been deemed suitable for publication in PLOS ONE. Congratulations! Your manuscript is now being handed over to our production team.

Kind regards, 

on behalf of

Dr. Michael R Hamblin 

Academic Editor

PLOS ONE